# SMALL INPUT NOISE IS ENOUGH TO DEFEND AGAINST QUERY-BASED BLACK-BOX ATTACKS

## ABSTRACT

While deep neural networks show unprecedented performance in various tasks, the vulnerability to adversarial examples hinders their deployment in safety-critical systems. Many studies have shown that attacks are also possible even in a black-box setting where an adversary cannot access the target model's internal information. Most black-box attacks are based on queries, each of which obtains the target model's output for an input, and many recent studies focus on reducing the number of required queries. In this paper, we pay attention to an implicit assumption of these attacks that the target model's output exactly corresponds to the query input. If some randomness is introduced into the model to break this assumption, query-based attacks may have tremendous difficulty in both gradient estimation and local search, which are the core of their attack process. From this motivation, we observe even a small additive input noise can neutralize most query-based attacks and name this simple yet effective approach Small Noise Defense (SND). We analyze how SND can defend against query-based black-box attacks and demonstrate its effectiveness against eight different state-of-the-art attacks with CIFAR-10 and ImageNet datasets. Even with strong defense ability, SND almost maintains the original clean accuracy and computational speed. SND is readily applicable to pre-trained models by adding only one line of code at the inference stage, so we hope that it will be used as a baseline of defense against query-based black-box attacks in the future.

## 1 INTRODUCTION

Although deep neural networks perform well in various areas, it is now well-known that small and malicious input perturbation can cause them to malfunction (Biggio et al., 2013; Szegedy et al., 2013). This vulnerability of AI models to adversarial examples hinders their deployment, especially in safety-critical areas. In a white-box setting, where the target model's parameters can be accessed, strong adversarial attacks such as Projected Gradient Descent (PGD) (Madry et al., 2018) can generate adversarial examples using the internal information. However, recent studies have shown that adversarial examples can be generated even in a black-box setting where the model's interior is hidden to adversaries.

These black-box attacks can be largely divided into *transfer-based attacks* and *query-based attacks*. Transfer-based attacks take advantage of *transferability* that adversarial examples generated from a network can deceive other networks. Papernot et al. (2017) train a substitute model that mimics the behavior of the target model and show that the adversarial example created from it can successfully disturb different models. However, due to differences in training methods and model architectures, the transferability of adversarial examples can be significantly weakened, and thus, transfer-based attacks usually result in lower success rates (Chen et al., 2017). For this reason, most black-box attacks are based on *queries*, each of which measures the target model's output for an input.

Query-based attacks create adversarial examples through an iterative process based on either local search with repetitive small input modifications or optimization with estimated gradients of an adversary's loss with respect to input. However, requesting many queries in their process takes a lot of time and financial loss. Moreover, many similar query images can be suspicious to system administrators. For this reason, researchers have focused on reducing the number of queries required to make a successful adversarial example.

Compared to the increasing number of studies on query-based attacks, the number of defenses against them is still very small (Bhambri et al., 2019). Also, existing defenses developed for white-box attacks may not be effective against query-based black-box attacks. Dong et al. (2020) find that existing defenses such as ensemble adversarial training (Tramèr et al., 2018) do not effectively defend against decision-based attacks. Therefore, it is necessary to develop new defense strategies that respond appropriately to the query-based attacks.

To defend against query-based black-box attacks, we pay attention to an implicit but important assumption of these attacks that the target model's output exactly corresponds to the query input. If some randomness is introduced into the model to break this assumption, they can have tremendous difficulty in both gradient estimation and local search, which are the core of their attack process. This intuition is illustrated in Fig. 1a.

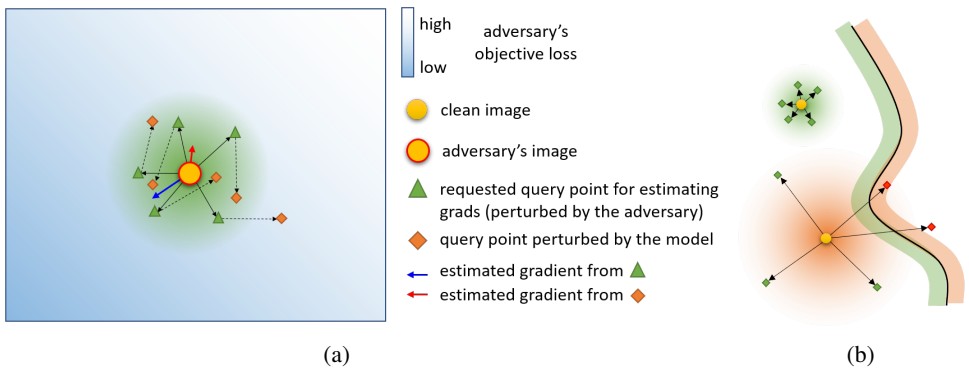

Figure 1: Illustrations of our intuitions. (a) Small noise can effectively disturb gradient estimation of query-based attacks which use finite difference (b) Compared to large noise, small noise hardly affects predictions on clean images.

Among previous studies, Dong et al. (2020) empirically find that randomization-based defenses are more effective in defending against query-based black-box attacks than other types of defenses. However, existing randomization-based defenses introduce much uncertainty into predictions, and thus, they also significantly degrade the accuracy of clean images (clean accuracy).

In this paper, however, we highlight that simply adding small Gaussian noise into an image is enough to defeat various query-based attacks by breaking the above core assumption while almost maintaining clean accuracy. One may think that additive Gaussian noise cannot defend against most adversarial attacks unless we introduce large randomness. This idea is valid for white-box attacks (Gu & Rigazio, 2014), but our experimental results show that small noise is surprisingly effective against query-based black-box attacks. Our second intuition regarding the minimization of clean accuracy loss can be seen in Fig. 1b. Dodge & Karam (2017) find that the classification accuracy decreases in proportion to the variance of Gaussian noise, but for a sufficiently small variance, the accuracy drop is negligible. Considering that the robustness against additive Gaussian noise is positively correlated to the distance to the decision boundary (Gilmer et al., 2019), the above observation implies that clean images have a relatively long distance to the decision boundary.

We think an adversarial defense should have the following goals: (1) preventing malfunction of a model against various attacks, (2) minimizing the computational overhead, (3) maintaining the accuracy on clean images, and (4) easily applicable to existing models. The proposed defense against query-based attacks meets all of the above objectives, and we name this simple yet effective defense *Small Noise Defense* (SND).

Our contributions can be listed as follows:

- We highlight the effectiveness of adding a small additive noise to input in defending against query-based black-box attacks. The proposed defense, SND, can be readily applied to pre-trained models by adding only one line of code in the Pytorch framework (Paszke et al., 2019) at the inference stage (`x = x + sigma * torch.randn_like(x)`) and almost maintains the performance of the model.

- We analyze how SND can efficiently interfere with gradient estimation and local search, which are the core of query-based attacks.

- We explain the difficulty of evading SND. We devise an adaptive attack against SND and explain its limitations in terms of query-efficiency.

- We have shown that the proposed method can effectively defend against eight different state-of-the-art query-based black-box attacks with the CIFAR-10 and ImageNet datasets. Specifically, we experimented with four decision-based and three score-based attacks in order to show strong defense ability against various attacks, including local search-based and optimization-based methods.

## 2 BACKGROUND

**Adversarial setting.** In this paper, we will deal with adversarial attacks on the image classification task. Suppose that a neural network $f(\boldsymbol{x})$ classifies an image $\boldsymbol{x}$ among total $N$ classes and returns a class-wise probability vector $\boldsymbol{y} = [y_1, ..., y_N]$ for $\boldsymbol{x}$. For notational convenience, we also denote the probability of $i^{th}$ class (i.e., $y_i$) as $f(\boldsymbol{x})_i$ and the top-1 class index as $h(\boldsymbol{x}) = \arg\max_{i \in C} y_i$, where $C = \{1, ..., N\}$.

In a black-box threat model, an adversary has a clean image $\boldsymbol{x}_0$ whose class index is $c_0$ and wants to generate an adversarial example $\hat{\boldsymbol{x}} = \boldsymbol{x}_0 + \boldsymbol{\delta}$ to fool a target model $f$. In the following, we denote the adversarial example at $t^{th}$ step in an iterative attack algorithm as $\hat{\boldsymbol{x}}_t$. The adversary should generate an adversarial example within a perturbation norm budget $\epsilon$ and query budget $Q$. If we let $q$ be the number of queries used to make $\boldsymbol{\delta}$, then we can write the adversary's objective as follows:

$$\min_{\boldsymbol{\delta}} \ell(\boldsymbol{x}_0 + \boldsymbol{\delta}), \text{ subject to } ||\boldsymbol{\delta}||_p \leq \epsilon \text{ and } q \leq Q, \tag{1}$$

where $\ell(\hat{\boldsymbol{x}}) = f(\hat{\boldsymbol{x}})_{c_0} - \max_{c \neq c_0} f(\hat{\boldsymbol{x}})_c$ for untargeted attacks and $\ell(\hat{\boldsymbol{x}}) = \max_{c \neq \hat{c}} f(\hat{\boldsymbol{x}})_c - f(\hat{\boldsymbol{x}})_{\hat{c}}$ for targeted attacks with target class index $\hat{c}$. Unless otherwise noted, in this paper, we use $p = 2$ and focus on untargeted attacks because it is more challenging for defenders. Besides, we assume that each pixel value is normalized into $[0, 1]$.

**Taxonomy of query-based black-box attacks.** Query-based attacks can be largely divided into score-based and decision-based according to the available type of the output of the target model (class-wise probabilities for score-based attacks and the top-1 class index for decision-based attacks). On the other hand, query-based attacks can be categorized into optimization-based attacks and local search-based attacks. Optimization-based methods optimize an adversary's objective loss with estimated gradients of the loss with respect to $\hat{\boldsymbol{x}}_t$. In contrast, local search-based attacks repeatedly update an image according to how the model's output changes after adding a small perturbation.

In the following, we briefly introduce various query-based attacks used in this paper.

**Bandit optimization with priors (Bandit-TD).** Ilyas et al. (2018) observe that the image gradients in successive steps of an iterative attack have strong correlation. In addition, they find that the gradients of surrounding pixels also have strong correlation. Bandit-TD exploits this information as priors for efficient gradient estimation.

**Simple Black-box Attack (SimBA & SimBA-DCT).** For each iteration, SimBA (Guo et al., 2019a) samples a vector $\boldsymbol{q}$ from a pre-defined set Q and modify the current image $\hat{\boldsymbol{x}}_t$ with $\hat{\boldsymbol{x}}_t - \boldsymbol{q}$ and $\hat{\boldsymbol{x}}_t + \boldsymbol{q}$ and updates the image in the direction of decreasing $\boldsymbol{y}_{c_0}$. Inspired by the observation that low-frequency components make a major contribution to misclassification (Guo et al., 2018), SimBA-DCT exploits DCT basis in low-frequency components for query-efficiency.

**Boundary Attack (BA).** BA (Brendel et al., 2018) updates $\hat{\boldsymbol{x}}_t$ on the decision-boundary so that the perturbation norm gradually decreases via random walks while misclassification is maintained.

**Sign-OPT.** Cheng et al. (2019a) treat a decision-based attack as a continuous optimization problem of the nearest distance to the decision boundary. They use the randomized gradient-free method (Nesterov & Spokoiny, 2017) for estimating the gradient of the distance. Cheng et al. (2019b) propose SIGN-OPT which uses the expectation of the sign of gradient with random directions to efficiently estimate the gradients without exhaustive binary searches.

**Hop Skip Jump Attack (HSJA).** Chen et al. (2020) improve Boundary Attack with gradient estimation. For each iteration of HSJA, it finds an image on the boundary with a binary search algorithm, and estimates the gradients, and calculates the step-size towards the decision boundary.

**GeoDA.** Rahmati et al. (2020) propose a geometry-based attack that exploits a geometric prior that the decision boundary of the neural network has a small curvature on average near data samples. By linearizing the decision boundary in the vicinity of samples, it can efficiently estimate the normal vector of the boundary, which helps to reduce the number of required queries for the adversarial attack.

## 2.1 Adversarial Defenses

As Dong et al. (2020) observe that randomization is important for effective defense against query-based attacks, we focus on randomization-based defenses among various defense methods. In what follows, we briefly explain three different randomization-based defenses along with PGD-adversarial training.

**Random Self-Ensemble (RSE).** RSE (Liu et al., 2018) adds Gaussian noise with $\sigma_{\text{inner}} = 0.1$ to the input of each convolutional layer, except for the first convolutional layer where $\sigma_{\text{init}} = 0.2$ is used. To stabilize the performance, they use an ensemble of multiple predictions for each image.

**Parametric Noise Injection (PNI).** He et al. (2019) propose a method to increase the robustness of neural networks by adding trainable Gaussian noise to activation or weight of each layer. They introduce learnable scale factors of noise and allow them to be learned with adversarial training.

**Random Resizing and Padding (R&P).** Xie et al. (2018) propose a random input transform-based method. In front of network inference, it applies random resizing and random padding to its input sequentially, making adversaries obtain noisy gradients. It can be easily applied to a pre-trained model, but it increases total computational time due to the enlarged input image.

**PGD-Adversarial Training (PGD-AT).** Madry et al. (2018) propose PGD-adversarial training which trains a model with adversarial examples generated by PGD attack. Unlike other defenses that are ineffective against adaptive attacks, it is well known that PGD-AT provides excellent defense against a variety of white-box attacks.

# 3 Analysis

## 3.1 Our approach

To defend against query-based black-box attacks, we add Gaussian noise with a sufficiently small $\sigma$ to the input as follows.

$$f_{\boldsymbol{\eta}}(\boldsymbol{x}) = f(\boldsymbol{x} + \boldsymbol{\eta}), \text{ where } \boldsymbol{\eta} \sim N(\boldsymbol{0}, \sigma^2 \boldsymbol{I}) \text{ and } \sigma \ll 1. \tag{2}$$

For an adversary, since the exact value of $\boldsymbol{\eta}$ is unknown, there can be multiple output values for any $\boldsymbol{x}$, so $f_{\boldsymbol{\eta}}$ is a random process. In what follows, we explain how this transform introduces tremendous difficulty in both gradient estimation and local search in query-based black-box attacks.

## 3.2 Defense Against Optimization-based Attacks

In this subsection, we will explain how small Gaussian input noise can disturb the gradient estimation in optimization-based attacks. We first look at defense against score-based attacks and then deal with decision-based attacks.

The core of optimization-based attacks is an accurate estimation of $\nabla \ell(\boldsymbol{x})$, which needs to be approximated with finite difference because of the black-box setting. For instance, the gradient can be estimated as $\tilde{\boldsymbol{g}}$ by Random Gradient-Free method (Nesterov & Spokoiny, 2017) as follows.

$$\tilde{\boldsymbol{g}} = \frac{1}{B} \sum_{i=0}^{B} \boldsymbol{g}_i, \text{ where } \boldsymbol{g}_i = \frac{\ell(\hat{\boldsymbol{x}}_t + \beta \boldsymbol{u}) - \ell(\hat{\boldsymbol{x}}_t)}{\beta} \boldsymbol{u} \text{ and } \boldsymbol{u} \sim N(\boldsymbol{0}, \sigma^2 \boldsymbol{I}). \tag{3}$$

Conceptually, by introducing small Gaussian noise into input, $\tilde{g}$ can differ greatly from the true gradient $\nabla \ell$ as shown in Fig. 1.

To illustrate it more formally, let us represent $\boldsymbol{\eta}$ by replacing it with $\boldsymbol{\eta}(\boldsymbol{x})$ to clarify $\boldsymbol{\eta}$ depends on both time and $\boldsymbol{x}$. Suppose $f^*_{\boldsymbol{\eta}(\boldsymbol{x})}$ is a sample function of the random process $f_{\boldsymbol{\eta}(\boldsymbol{x})}(\boldsymbol{x})$ at some time. Then, this function is very noisy because of $\boldsymbol{\eta}(\boldsymbol{x})$ with regard to $\boldsymbol{x}$. We also assume that $\ell^*$ is derived from $f^*_{\boldsymbol{\eta}(\boldsymbol{x})}$, then unless $\mathrm{Var}[\ell^*(\boldsymbol{x} + \boldsymbol{u})]$ is near zero, $\ell^*$ is discontinuous and non-differentible and thus, $\nabla \ell^*$ does not exist. Therefore, the gradient estimation using finite differences does not converge to the target gradient $\nabla \ell$.

In decision-based attacks, $\hat{\boldsymbol{x}}_t$ is likely to be in the vicinity of the decision boundary. Therefore even the small noise can move $\hat{\boldsymbol{x}}_t$ across the boundary so that the output is changed. The estimated gradient through frequently erroneous predictions hinders the generation of adversarial examples. We illustrate the above defense mechanism in Appendix A. On the other hand, the error of the binary search algorithm, which is widely used to calculate the distance to the decision boundary, can be amplified due to $\boldsymbol{\eta}$. Therefore, algorithms such as HSJA, which assume that $\hat{\boldsymbol{x}}$ is near the decision boundary, are likely to work incorrectly.

### 3.3 DEFENSE AGAINST LOCAL SEARCH-BASED ATTACKS

Almost all local search-based attacks do not take into account the uncertainty of the model output. Suppose an adversary recognizes that the attack objective loss decreases for $\boldsymbol{x} + \boldsymbol{\tau}$ where $\boldsymbol{\tau}$ is a perturbation, and updates $\boldsymbol{x}$ as $\boldsymbol{x} + \boldsymbol{\tau}$. However, since the actual evaluated input of $f$ is $\boldsymbol{x} + \boldsymbol{\tau} + \boldsymbol{\eta}$ where $\boldsymbol{\eta} \sim N(\boldsymbol{0}, \sigma^2 \boldsymbol{I})$, the attack objective loss might be rather increases at the originally intended input $\boldsymbol{x} + \boldsymbol{\tau}$. This prediction error makes the attack algorithm stuck in the iterative process and prevents generating adversarial examples.

### 3.4 DIFFICULTY OF EVADING SMALL NOISE DEFENSE

Although the input of the function, $\boldsymbol{x} + \boldsymbol{\eta}$, is a Gaussian random process, but $f(\boldsymbol{x} + \boldsymbol{\eta})$, the result of the nonlinear function $f$, is no longer a Gaussian random process. This makes it very difficult for query-based attacks to bypass SND. One can approximate $f(\boldsymbol{x})$ as $\mathbb{E}_{\boldsymbol{\eta}}[f_{\boldsymbol{\eta}}(\boldsymbol{x})]$ by taking the expectation through multiple queries using the fact that $\mathbb{E}(\boldsymbol{\eta}) = \boldsymbol{0}$. However, this attempt requires many queries for each iteration and greatly diminishes query efficiency. In addition, even if a large amount of queries are used, $\mathbb{E}_{\boldsymbol{\eta}}[f_{\boldsymbol{\eta}}(\boldsymbol{x})]$ may be different from $f(\boldsymbol{x})$ because of the nonlinearity of the deep neural networks. With a simple example, we explain how the expectation value differs from the actual value when Gaussian noise is added to input of nonlinear functions.

Let $F(\boldsymbol{x}) = \boldsymbol{x}^T \boldsymbol{x}$ where $F : \mathbb{R}^d \to \mathbb{R}$, and $F_{\boldsymbol{\eta}}(\boldsymbol{x}) = F(\boldsymbol{x} + \boldsymbol{\eta})$ where $\boldsymbol{\eta} \sim N(\boldsymbol{0}, \sigma^2 \boldsymbol{I})$. Suppose we want to estimate $F(\boldsymbol{0})$ with $F_{\boldsymbol{\eta}}(\boldsymbol{0})$. Then, $\mathbb{E}[F_{\boldsymbol{\eta}}(\boldsymbol{0})] = \mathbb{E}[(\boldsymbol{0} + \boldsymbol{\eta})^T(\boldsymbol{0} + \boldsymbol{\eta})] = \mathbb{E}[\boldsymbol{\eta}^T \boldsymbol{\eta}] = d\sigma^2$. Therefore, $\mathbb{E}[F_{\boldsymbol{\eta}}(\boldsymbol{0})] = d\sigma^2 \neq 0 = F(\boldsymbol{0})$ and if $d$ is very large (e.g., for an image of size 224×224×3, $d=150,528$), then the estimation error would be high.

Additionally, in the case of a simple network consisting of an affine layer and ReLU activation, $F(x) = \mathrm{ReLU}(Wx + b)$ and $F : \mathbb{R} \to \mathbb{R}$. Since $F(x)$ is $\max(0, Wx + b)$ and $\mathbb{E}[F_{\eta}(x)]$ is $Wx + b - \Phi(-\frac{Wx+b}{|W|\sigma}) + |W|\sigma\phi(-\frac{Wx+b}{|W|\sigma})$, the estimation error would exist (The detailed proof is in Appendix E). From the proof on the simple network, we can expect that the average of the output may have an error with the actual output even in a deep neural network.

## 4 EXPERIMENTS AND DISCUSSION

### 4.1 EXPERIMENTAL SETTINGS

In this section, we evaluate the defense ability of SND against eight different query-based black-box attacks: BA, Sign-OPT, HSJA, GeoDA, SimBA, SimBA-DCT, Bandit-TD, Subspace Attack along with other defense methods: PNI, RSE, R&P, PGD-AT. We use the CIFAR-10 (Krizhevsky et al., 2009) and ImageNet (Deng et al., 2009) datasets for our experiments and following previous studies (Chen et al., 2020; Guo et al., 2019a; He et al., 2019), we use ResNet-20 for CIFAR-10 and ResNet-50 (He et al., 2016) for ImageNet for target networks. Following Brendel et al. (2018), we randomly

sampled 1,000 and 250 correctly classified images from the CIFAR-10 test set and the ImageNet Validation set for evaluation. We describe a detailed experimental setting in Appendix B.

For evaluation metrics, we first define a *successfully attacked image* as an image from which an attack can find an adversarial image within the perturbation budget $\epsilon$ and query budget $Q$. With this definition, we use *attack success rate*, which is the percentage of the number of successfully attacked images over the total number of evaluated images. Note that since we evaluate defense performance, **a lower attack success rate is better**. We use $\epsilon = 1.0$ for the CIFAR-10 dataset, and $\epsilon = 5.0$ for the ImageNet dataset. In addition, we denote the $q^{th}$ query image as $\hat{x}^q$. Note that $\hat{x}^q$ and $\hat{x}_t$ can be different.

## 4.2 EXPERIMENTAL RESULTS

**Evaluation of clean accuracy.** We first evaluate the clean accuracy of models with defenses on the original test split (10k images) of the CIFAR-10 and validation split (50k images) of the ImageNet dataset. As shown in Table 1, SND hardly reduces the clean accuracy compared to other methods. The accuracy drop caused by SND is not significant at $\sigma \leq 0.01$ and becomes large at $\sigma = 0.02$, which implies that sufficiently small $\sigma$ is important for maintaining clean accuracy.

Table 1: Comparison of clean accuracy. For randomization-based methods, we denote mean and standard deviation of clean accuracy in 5 repetitive experiments with different random seeds.

| ResNet-20 on CIFAR-10 | | ResNet-50 on ImageNet | |
|---|---|---|---|
| Defense | Clean Accuracy (%) | Defense | Clean Accuracy (%) |
| Baseline | 91.34 | Baseline | 76.13 |
| SND ($\sigma = 0.001$) | $91.33 \pm 0.02$ | SND ($\sigma = 0.001$) | $76.10 \pm 0.02$ |
| SND ($\sigma = 0.01$) | $90.57 \pm 0.09$ | SND ($\sigma = 0.01$) | $75.47 \pm 0.03$ |
| SND ($\sigma = 0.02$) | $87.56 \pm 0.18$ | SND ($\sigma = 0.02$) | $73.91 \pm 0.02$ |
| RSE | $83.40 \pm 0.15$ | PGD-AT | 57.9 |
| PNI | $85.15 \pm 0.18$ | R&P | $74.26 \pm 0.07$ |

**Evaluation on the CIFAR-10 dataset.** We performed four different decision-based attacks against models with defenses, and Table 2 shows the evaluated attack success rates. SND shows competitive defense ability despite having more than 5% higher clean accuracy compared to other defenses. Moreover, due to significant performance drop, RSE and PNI cannot be applied to the models for large-scale image classification with the ImageNet dataset.

Table 2: Evaluation of attack success rates (%) on the CIFAR-10 dataset.

| Attack method | BA | | | Sign-OPT | | | HSJA | | | GeoDA | | |
|---|---|---|---|---|---|---|---|---|---|---|---|---|
| # of queries | 2k | 5k | 10k | 2k | 5k | 10k | 2k | 5k | 10k | 2k | 5k | 10k |
| Baseline | 36.2 | 69.5 | 84.6 | 59.1 | 88.9 | 91.2 | 86.4 | 89.2 | 89.2 | 64.7 | 71.3 | 76.5 |
| SND ($\sigma = 0.01$) | 14.0 | 17.8 | 20.0 | 21.7 | 22.3 | 22.8 | 16.5 | 19.9 | 22.7 | 12.0 | 12.1 | 12.4 |
| SND ($\sigma = 0.001$) | 33.0 | 53.0 | 61.3 | 20.6 | 22.2 | 23.4 | 48.1 | 67.6 | 81.9 | 11.7 | 11.8 | 12.2 |
| RSE | 18.2 | 19.0 | 19.8 | 18.7 | 18.7 | 18.7 | 19.9 | 22.2 | 23.4 | 19.7 | 19.9 | 20.5 |
| PNI | 15.1 | 15.4 | 15.8 | 18.1 | 18.1 | 18.1 | 17.4 | 19.2 | 20.6 | 19.9 | 20.2 | 20.4 |

**Evaluation on the ImageNet dataset.** We performed six different query-based attacks against models with defenses, and Table 3 shows the evaluated attack success rates. When the query budget $Q$ is 20k, the average of the attack success rates over the attacks against the baseline is 87.9%, whereas SND with $\sigma = 0.01$ significantly reduces it to 10.0%. SND with $\sigma = 0.001$ also significantly reduces the average attack success rate to 29.5%, which is comparable to the second-best method, R&P (22.5%). We also calculate the average $\ell_2$ norm of adversarial perturbations $||x_0 - \hat{x}^q||_2$ at the predefined query budget $Q$ to show whether the perturbation norm diverges or not. If an attack stops in the middle without requesting all $Q$ queries, we use the last query image instead. In decision-based attacks, it can be seen that randomization-based defenses, SND and R&P, significantly increase the perturbation norm as $q$ increases. In SimBA and SimBA-DCT, the perturbation norm is minimal in SND and R&P, which implies that the attacks have significant difficulty in finding a perturbation which decreases $y_{c_0}$.

Table 3: Evaluation of attack success rates against defenses on the ImageNet dataset. We denote the average $\ell_2$ norm of perturbations in the parenthesis.

| Attack type | Decision-based Attack | | | | | | | | |
|---|---|---|---|---|---|---|---|---|---|
| Attack method | Sign-OPT | | | HSJA | | | GeoDA | | |
| # of queries | 5k | 10k | 20k | 5k | 10k | 20k | 5k | 10k | 20k |
| Baseline | 36.4% | 62.4% | 88.0% | 64.0% | 88.4% | 99.6% | 50.0% | 62.8% | 72.0% |
| | (9.71) | (4.72) | (2.38) | (4.43) | (2.34) | (1.28) | (6.38) | (5.04) | (4.12) |
| SND ($\sigma$=0.01) | **6.8%** | **6.8%** | **7.2%** | **6.4%** | **7.6%** | **8.4%** | **7.2%** | **7.6%** | **7.6%** |
| | (41.37) | (70.27) | (87.94) | (31.03) | (26.81) | (22.93) | (33.48) | (33.14) | (32.61) |
| SND ($\sigma$=0.001) | **6.8%** | 7.6% | 8.0% | 13.6% | 20.4% | 32.4% | 8.4% | 8.8% | 9.2% |
| | (35.15) | (54.33) | (88.29) | (11.60) | (8.86) | (6.75) | (27.44) | (25.88) | (24.41) |
| AT | 28.8% | 30.0% | 32.4% | 30.4% | 33.2% | 36.0% | 32.4% | 34.0% | 35.6% |
| | (30.67) | (24.66) | (19.72) | (20.99) | (17.27) | (13.46) | (14.54) | (13.63) | (12.90) |
| R&P | 13.2% | 13.2% | 13.2% | 13.6% | 15.2% | 16.0% | 14.4% | 14.4% | 15.2% |
| | (51.19) | (82.14) | (85.78) | (33.01) | (31.00) | (29.42) | (31.72) | (31.21) | (30.45) |
| Attack type | Score-based Attack | | | | | | | | |
| Attack method | SimBA | | | SimBA-DCT | | | Bandit-TD | | |
| # of queries | 5k | 10k | 20k | 5k | 10k | 20k | 5k | 10k | 20k |
| Baseline | 74.0% | 74.4% | 74.4% | 94.8% | 95.2% | 95.2% | 94.0% | 97.2% | 98.4% |
| | (3.89) | (3.99) | (4.02) | (3.12) | (3.14) | (3.14) | (4.70) | (4.70) | (4.70) |
| SND ($\sigma$=0.01) | **8.4%** | **9.2%** | **10.0%** | **8.4%** | **8.8%** | **10.4%** | 15.2% | 15.2% | 16.4% |
| | (0.52) | (0.55) | (0.57) | (0.56) | (0.58) | (0.60) | (4.74) | (4.74) | (4.74) |
| SND ($\sigma$=0.001) | 27.2% | 35.6% | 50.4% | 46.4% | 60.4% | 68.4% | **7.2%** | **7.6%** | **8.4%** |
| | (1.84) | (2.14) | (2.43) | (2.22) | (2.44) | (2.58) | (4.83) | (4.83) | (4.83) |
| AT | 27.6% | 27.6% | 27.6% | 36.0% | 36.0% | 36.0% | 38.8% | 45.2% | 52.8% |
| | (5.46) | (7.55) | (10.17) | (5.36) | (6.21) | (6.62) | (3.54) | (3.54) | (3.54) |
| R&P | 26.4% | 27.6% | 28.0% | 27.2% | 28.4% | 29.2% | 32.0% | 33.2% | 33.6% |
| | (0.48) | (0.51) | (0.54) | (0.52) | (0.55) | (0.58) | (4.50) | (4.50) | (4.50) |

**Empirical evidence for assumptions of SND.** To provide the supporting evidence for assumptions of SND in score-based attacks, we calculate $\hat{\sigma} = \sqrt{\mathrm{Var}[\ell(\boldsymbol{x} + \boldsymbol{\eta})]}$ where $\boldsymbol{\eta} \sim N(\boldsymbol{0}, \sigma^2 \boldsymbol{I})$. With $\sigma = 0.01$ and 1,000 test images of the CIFAR-10, we evaluate $\ell(\boldsymbol{x} + \boldsymbol{\eta})$ for 100 iterations for each clean image and calculate the $\hat{\sigma}$ averaged over all images. In our experiment, $\hat{\sigma} = 0.04$ which is small but sufficient to make $\ell(\boldsymbol{x} + \boldsymbol{\eta})$ to be non-differentiable about $\boldsymbol{x}$.

In decision-based attacks, our assumption in Section 3.2 is that if $\hat{\boldsymbol{x}}$ is near the decision boundary, sufficiently small noise can easily move the image across the boundary. We evaluate $P_{mis} := P(h(\boldsymbol{x}) \neq h(\boldsymbol{x} + \boldsymbol{\eta}))$ through experiments. We count the above mismatch case for all queries during the attack process. With $\sigma = 0.001, 0.01$ and the CIFAR-10 test images, the average $P_{mis}$ for all attacks is calculated as 0.22 and 0.25, respectively. In constrast, on clean images, $P_{mis}$ is obtained as 0.002 and 0.021 respectively. Therefore, the results shown in Table 5 support our argument.

**Evaluation of an adaptive attack against SND.** As described in Section 3.4, we devise an adaptive attack against SND that takes the expectation of predictions for repetitive $T$ queries. In this experiment, we perform HSJA against SND with $\sigma = 0.01$ on the CIFAR-10 dataset. Since HSJA is a decision-based attack, we regard the most predicted class in $T$ queries as the expected class. We measure the attack success rate and $P_{mis}$ according to the query budget, and the adaptive attack clearly shows a higher attack success rate than the baseline ($T = 1$), as shown in Table 4. On the same query budget, however, the adaptive attack shows a lower attack success rate (e.g., 22.7% ($T$=1) > 18.2% ($T$=5) at $Q$=10k and 29.3% ($T$=5) > 27.3% ($T$=10) at $Q$=50k). Therefore, the expectation-based adaptive attack has limitations due to the restricted query budget. Moreover, even if $T$ increases, the $P_{mis}$ does not decrease and this reinforces our argument in Section 3.4. We also apply the adaptive attack with $T$=10 to BA, SO, and GeoDA for comparison. The results are shown in Table 6.

**Varying $\sigma$ for each inference.** So far, we have used a fixed $\sigma$ for SND. Changing $\sigma$ for each query may reduce clean accuracy while maintaining the defense ability. From this motivation, we multiply $\boldsymbol{\eta}$ with $k$ which is randomly determined between 0 and 1 using the beta distribution with three

Table 4: Evalutation of SND with different $T$.

| # of queries | 2k×$T$ | 5k×$T$ | 10k×$T$ | $P_{mis}$ |
|---|---|---|---|---|
| HSJA ($T$=1) | 16.5% | 19.9% | 22.7% | 0.189 |
| HSJA ($T$=5) | 18.2% | 23.8% | 29.3% | 0.321 |
| HSJA ($T$=10) | 21.0% | 27.3% | 34.9% | 0.376 |
| HSJA ($T$=20) | 25.2% | 34.0% | 46.6% | 0.410 |
| # of queries | 50k | 100k | 200k | |
| HSJA | 29.0% | 30.1% | 31.0% | 0.120 |

Table 5: Evaluation of $P(h(\boldsymbol{x}) \neq h(\boldsymbol{x} + \boldsymbol{\eta}))$.

| Defense Attack | SND ($\sigma = 0.001$) | SND ($\sigma = 0.01$) |
|---|---|---|
| BA | 0.134 | 0.227 |
| Sign-OPT | 0.216 | 0.215 |
| HSJA | 0.255 | 0.189 |
| GeoDA | 0.314 | 0.391 |
| None | 0.002 | 0.021 |

Table 6: Evalutation of the adaptive attack against SND with $T$=10.

| # of queries | 2k×$T$ | 5k×$T$ | 10k×$T$ | $P_{mis}$ |
|---|---|---|---|---|
| BA ($T$=10) | 20.3% | 28.5% | 32.9% | 0.413 |
| SO ($T$=10) | 20.5% | 21.1% | 21.9% | 0.354 |
| HSJA ($T$=10) | 21.0% | 27.3% | 34.9% | 0.376 |
| GeoDA ($T$=10) | 12.1% | 12.2% | 12.2% | 0.413 |

different ways: (1) Uniformly random (the same as $\alpha$=$\beta$=1) (2) Sampling from a beta distribution with $\alpha$=$\beta$=2 whose probability density function (PDF) is ∩-shaped. (3) Sampling from a beta distribution with $\alpha$=$\beta$=0.5 whose PDF is ∪-shaped. We calculate clean accuracy and average $\ell_2$ norm of perturbations for each method. Among the three ways, $\alpha$=$\beta$=2 is better than the others, but SND with fixed $\sigma$=0.01 is better than SND with variable $\sigma$, which implies large randomness in each query is crucial for a strong defense. Detailed results can be found in Appendix C.

**Defense against hybrid black-box attack.** Recently proposed Subspace attack (Guo et al., 2019b) exploits transferability-based priors, gradients from local substitute models trained on a small proxy dataset. We use pre-trained ResNet-18 and ResNet-34 (He et al., 2016) as reference models for gradient priors. For this attack, we perform an attack based on $\ell_\infty$ norm because the authors provide parameter settings only for $\ell_\infty$ norm. Detailed results are described in Appendix D, but the result shows that SND alone cannot effectively defend against the hybrid attack with gradient priors. However, when SND is combined with PGD-AT, it effectively protects the model and decreases the attack success rate from 100% to 42.4% at $Q$=20k and $\sigma$=0.01. To focus on the defensive ability against gradient estimation, we recalculate the attack success rate without initially misclassified images. Then, the newly obtained attack success rate decreases from 100% to 16.4% at $Q$=20k. This result implies that SND can be combined with other defenses against transfer-based attacks to achieve strong defense ability against all types of black-box attacks.

## 5 RELATED WORK

**History-based detection methods against query-based black-box attacks.** To the best of our knowledge, studies that mainly target defending against query-based black-box attacks have not yet been published. However, history-based detection techniques for query-based attacks have been proposed recently (Chen et al., 2019; Li et al., 2020). Considering that adversary requires many queries of similar images for finding an adversarial example, they store information about past query images to detect the unusual behavior of query-based attacks.

**Certified defense with additive Gaussian noise.** Li et al. (2019) analyze the connection between robustness of models against additive Gaussian noise and adversarial perturbations. They derive the certified bounds on the norm bounded adversarial perturbation and they propose a new training strategy to improve the certified robustness. Simlilarly, randomized smoothing (Cohen et al., 2019) creates a smoothed classifier that correctly classifies when Gaussian noise is added to the classifier's input. Cohen et al. (2019) prove that this smoothed classifier can have $\ell_2$ certified robustness for an input. Both SND and the above certified defenses add Gaussian noise to the input. However, the purpose of the addition of noise in the certified defenses is to induce the classifier to gain certified robustness. Whereas, SND adds noise to disturb an accurate measurement of the output to defend

against query-based black-box attacks at the inference. In addition, the certified defenses use a much larger $\sigma$ ($\geq 0.25$) than SND (0.01).

## 6 CONCLUSION

In this paper, we highlight that even a small additive input noise can effectively neutralize query-based black-box attacks and name this approach Small Noise Defense (SND). We demonstrate its effectiveness against eight different query-based attacks with CIFAR-10 and ImageNet datasets. Our work suggests that query-based black-box attacks should consider the randomness of the target network as well. SND is readily applicable to pre-trained models by adding only one code line without any structural changes and fine-tuning. Due to its simplicity and effectiveness, we hope that SND will be used as a baseline of defense against query-based black-box attacks in the future.

We can list various exemplary future work as follows: (1) Finding a better type of noise. Other types of noise, such as salt-and-pepper or image processing like random contrast, can be used instead of Gaussian noise as long as it can disrupt query-based attacks while maintaining clean accuracy; (2) Devising effective black-box attacks against SND while maintaining query efficiency. The uncertainty of randomized predictions can be mitigated through expectation, but it requires many queries and hurts query-efficiency; (3) Rigorous proof of the defense ability of small input noise. We explain how SND can defend against query-based attacks, but one can establish theorems like the lower bound of mean squared error of estimated gradient with small input noise.

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

# A  ILLUSTRATION OF DEFENSE AGAINST DECISION-BASED ATTACK WITH SMALL INPUT NOISE

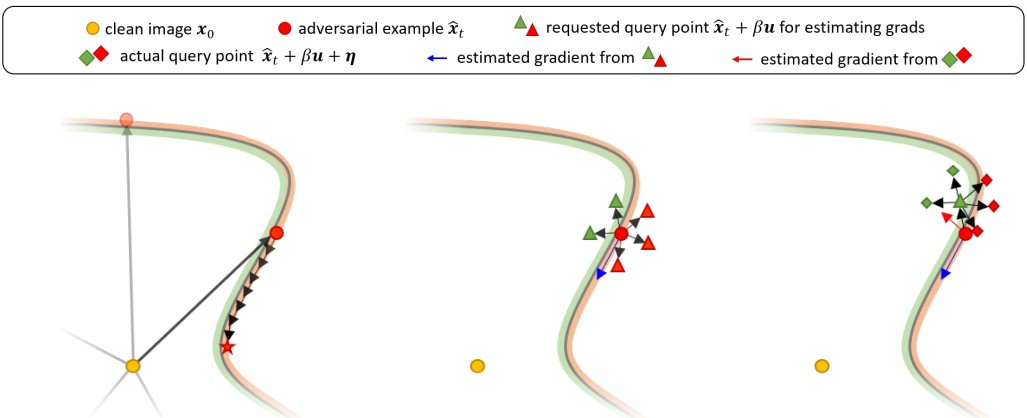

Figure 2: Illustration of how small noise can defend against decision-based attacks (left) An adversary wants to reach the optimal adversarial example from an initial misclassifed point. (middle) To find the direction of the next update, it evaluates $\hat{x}_t + u$ (right) Small noise can disturb this gradient estimation.

# B  DETAILED EXPERIMENTAL SETTING

## B.1  SETTINGS OF ATTACK METHODS

**BA:** We adopt BA provided by Adversarial Robustness Toolbox (ART) library (Nicolae et al., 2018) with default parameters.

**Sign-OPT:** We port the code[1] provided by the authors into our framework without changing the special parameters of the attack.

**HSJA:** We adopt HSJA provided by Adversarial Robustness Toolbox (ART) library with default parameters except for increasing the maximum number of iterations to 64 to follow the authors' code[2].

**GeoDA:** We port the code[3] provided by the authors into our framework without changing the special parameters of the attack. **SimBA & SimBA-DCT:** We port the code[4] provided by the authors into our framework. Following the authors, we use `freq_dims`=28, `order`=strided, and `stride`=7 for SimBA-DCT.

**Bandit-TD:** We port the code[5] provided by the authors into our framework with default parameters except for `batch_size`=1 and `epsilon`=4.9.

**Subspace Attack:** We port the code[6] provided by the authors into our framework with the original setting for $\ell_\infty$ norm untargeted attack for the ImageNet. We use pre-trained ResNet-18 and ResNet-34 trained on the `imagenetv2-val` dataset as reference models that are provided by the authors.

---

[1]https://github.com/cmhcbb/attackbox
[2]https://github.com/Jianbo-Lab/HSJA
[3]https://github.com/thisisalirah/GeoDA
[4]https://github.com/cg563/simple-blackbox-attack
[5]https://github.com/MadryLab/blackbox-bandits
[6]https://github.com/ZiangYan/subspace-attack.pytorch

## B.2 SETTINGS OF DEFENSE METHODS

**Baseline:** We train a ResNet-20 model on the CIFAR-10 dataset for 200 epochs and use this model for our experiments. For the ImageNet dataset, we use the pre-trained ResNet-50 model provided by the Pytorch library.

**PNI:** We use the pre-trained ResNet-20 model with PNI-W (channel-wise) provided by the authors trained on the CIFAR-10 dataset.

**PGD-AT:** We use the adversarilly trained ResNet-50 model for $\ell_2$ norm with $\epsilon_{train} = 3$ provided from *robustness* library (Engstrom et al., 2019) with PGD on the ImageNet dataset for comparison.

**RSE:** We train a RSE-based ResNet-20 with $\sigma_{\text{init}} = 0.2$ and $\sigma_{\text{inner}} = 0.1$. Considering computational efficiency, we use 5 ensembles for each prediction of RSE.

**R&P:** R&P applies random resizing and random padding to its input sequentially. It first rescales the input image of size $W \times H \times 3$ with a scale factor $s$ which is sampled from $[s_{min}, s_{max}]$, and places it in a random position within an empty image of size $s_{max}W \times s_{max}Y \times 3$. Following the authors, we set $s_{min}$ and $s_{max}$ as $\frac{310}{299}$ and $\frac{331}{299}$ respectively.

## C  EVALUATION OF VARYING $\sigma$ FOR EACH INFERENCE

Table 7: Experimental results of varying $\sigma$ with the CIFAR-10 dataset. We evaluate mean and standard deviation of clean accuracy in 5 repetitive experiments on the original test dataset.

| | Clean Acc. (%) | $\mathbb{E}_q\|\boldsymbol{x}_0 - \hat{\boldsymbol{x}}^q\|_2$ | Sign-OPT | | | HSJA | | |
|---|---|---|---|---|---|---|---|---|
| # of queries | - | - | 2k | 5k | 10k | 2k | 5k | 10k |
| $\sigma$=0.001 | $91.33 \pm 0.02$ | 0.055 | 20.6% | 22.2% | 23.4% | 48.1% | 67.6% | 81.9% |
| $\sigma$=0.01 | $90.57 \pm 0.09$ | 0.550 | 21.7% | 22.3% | 22.8% | 16.5% | 19.9% | 22.7% |
| $\sigma$=0.01, $\alpha=\beta$=1 | $91.04 \pm 0.12$ | 0.276 | 21.6% | 22.4% | 22.9% | 18.1% | 23.9% | 30.0% |
| $\sigma$=0.01, $\alpha=\beta$=2 | $91.15 \pm 0.04$ | 0.275 | 20.3% | 20.8% | 21.7% | 19.7% | 23.5% | 28.5% |
| $\sigma$=0.01, $\alpha=\beta$=0.5 | $91.06 \pm 0.06$ | 0.275 | 20.9% | 21.4% | 22.4% | 19.8% | 26.3% | 32.6% |
| $\sigma$=0.02 | $87.56 \pm 0.18$ | 1.098 | 26.4% | 26.5% | 26.6% | 24.2% | 26.7% | 30.1% |
| $\sigma$=0.02, $\alpha=\beta$=1 | $90.17 \pm 0.09$ | 0.552 | 22.0% | 22.0% | 22.1% | 19.6% | 23.2% | 25.4% |
| $\sigma$=0.02, $\alpha=\beta$=2 | $90.44 \pm 0.08$ | 0.550 | 21.6% | 21.8% | 22.1% | 18.0% | 22.1% | 25.0% |
| $\sigma$=0.02, $\alpha=\beta$=0.5 | $89.99 \pm 0.24$ | 0.549 | 22.5% | 22.5% | 23.3% | 20.3% | 24.4% | 27.5% |

## D  EVALUATION OF ATTACK SUCCESS RATES OF SUBSPACE ATTACK

Table 8: Evaluation of attack success rates of Subspace Attack against defenses on the ImageNet dataset. We also calculate the attack success rate without initially misclassified images and denote it in the parenthesis.

| Attack method | Subspace Attack | | |
|---|---|---|---|
| # of queries | 5k | 10k | 20k |
| Baseline | 99.6% (99.6%) | 100.0% (100.0%) | 100.0% (100.0%) |
| SND ($\sigma = 0.01$) | 61.2% (59.6%) | 61.2% (59.6%) | 61.2% (59.6%) |
| SND ($\sigma = 0.001$) | 64.4% (64.4%) | 66.4% (66.4%) | 68.4% (68.4%) |
| PGD-AT | 71.6% (45.6%) | 78.4% (52.4%) | 81.6% (55.6%) |
| PGD-AT + SND ($\sigma = 0.01$) | 40.8% (14.8%) | 42.0% (16.0%) | 42.4% (16.4%) |
| PGD-AT + SND ($\sigma = 0.001$) | 62.0% (36.0%) | 62.4% (36.4%) | 63.2% (37.2%) |
| R&P | 73.2% (68.4%) | 74.0% (69.2%) | 74.0% (69.2%) |

# E   THE PROOF OF THE EXPECTATION ERROR ON A SIMPLE NETWORK

Let $F(x) = \text{ReLU}(Wx + b)$ where $F : \mathbb{R} \to \mathbb{R}$ and $\text{ReLU}(x) = \max(0, x)$, and $F_\eta(x) = F(x + \eta)$, where $\eta \sim N(0, \sigma^2)$. Suppose we want to estimate $F(x)$ with $\mathbb{E}[F_\eta(x)]$. Let $W(x + \eta) + b$ be $Y$, then $Y$ can be represented with $\mu_y = Wx + b$ and $\sigma_y^2 = W^2 \sigma^2$ as:

$$Y \sim N(\mu_y, \sigma_y^2). \tag{4}$$

Then, $F_\eta(x)$ is $\max(0, Y)$ and $\mathbb{E}[F_\eta(x)] = \mathbb{E}[\max(0, Y)]$ can be obtained by the law of total expectation.

$$\begin{aligned}
\mathbb{E}[F_\eta(x)] &= \mathbb{E}[\max(0, Y)] \\
&= \mathbb{E}[Y|Y > 0]\Pr(Y > 0) + 0\Pr(Y \le 0)
\end{aligned} \tag{5}$$

Using the truncated normal distribution, we recall the fact as follows:

$$\mathbb{E}[Y|Y > a] = \mu_y + \sigma_y \frac{\phi((a - \mu_y)/\sigma_y)}{1 - \Phi((a - \mu_y)/\sigma_y)}, \tag{6}$$

where $\phi(x) = \frac{1}{\sqrt{2\pi}}\exp(-\frac{1}{2}x^2)$ and $\Phi$ is the cumulative distribution function of the standard normal distribution. Since $\Pr(Y > 0)) = 1 - \Phi(\frac{\mu_y}{\sigma_y})$, $\mathbb{E}[F_\eta(x)]$ can be represented as:

$$\begin{aligned}
\mathbb{E}[F_\eta(x)] &= \mu_y(1 - \Phi(-\frac{\mu_y}{\sigma_y})) + \sigma_y \phi(-\frac{\mu_y}{\sigma_y}) \\
&= Wx + b - \Phi(-\frac{Wx + b}{|W|\sigma}) + |W|\sigma\phi(-\frac{Wx + b}{|W|\sigma}).
\end{aligned} \tag{7}$$

