# OpenReview forum: "Small Input Noise is Enough to Defend Against Query-based Black-box Attacks"
_ICLR.cc/2021/Conference — Reject_

### Official Review · AnonReviewer3 · 2020-10-27
**Official Blind Review #3**

**Rating:** 3
**Confidence:** 5

**Review:**

This paper proposes an adversarial defense method against black-box attack by adding random noise on adversarial examples. The method is simple and empirical results verify that the proposed defense can be used to decrease the attack success rate of both optimization-based and local search-based attacks.

My main concern of this paper is the simply use of randomness for adversarial defense, rather than directly improving the robustness of the target model itself (e.g. adversarial training). In the discussion about the Obfuscated Gradient [1], the authors criticized the randomness-based defense method: “Stochastic Gradients are caused by randomized defenses, where either the network itself is randomized or the input is randomized before being fed to the classifier.” Although this paper is aimed at defending against black-box attacks, it is still essentially a defense by adding random noise to the input data. In addition, the proposed method is not useful to defense against white-box attacks and transfer-based attacks, since basically it does not change the predictive model.

Another problem is that while the defense mechanism in this paper is simple, there are equally simple ways to circumvent it. An example is that when the randomness of model output is found by the adversary based on Boundary Attack, the attacker only needs to change the judgment condition of successful attack to “most of the attacks are successful in the same input for multiple queries” to avoid the decrease of attack success rate.

[1] https://arxiv.org/abs/1802.00420

---

> ### Public Comment · ~Nicholas_Carlini1 · 2020-11-12
> **Comment on"obfuscated gradients" quotation**
>
> To be fair to this paper (as an author of [1], above) our criticism of randomness was intended to discourage defenses to say "we add randomness, and now there's so much randomness that the attacker can't guess what we picked, therefore we're secure". This paper does correctly try to take ensembles over the random values in order to break the defense, and in that sense is respecting our advice.
>
> That said I agree with R3 that the correct way to evaluate robustness is to try and construct new attacks that break the proposed defense. If taking an average over random samples is sufficient is up to the reviewers.

---

> > ### Author Response · Authors · 2020-11-13
> > **We thank Carlini for taking an interest in our work**
> >
> > We thank Carlini for taking an interest in our work and the valuable comment.
> > We totally agree with the advice in [2] that correct evaluation of an adversarial defense requires testing against adaptive attacks that exactly know the working principle of the defense. Following the suggestion in [1], we design and experiment with expectation-based adaptive attacks against SND. These adaptive attacks can be seen as an adaptation of Expectation Over Transform (EOT) [3] technique to the gradient estimation of query-based attack.
> >
> > [1] Athalye, Anish, Nicholas Carlini, and David A. Wagner. "Obfuscated Gradients Give a False Sense of Security: Circumventing Defenses to Adversarial Examples." ICML. 2018.\
> > [2] Carlini, Nicholas, et al. "On evaluating adversarial robustness." arXiv preprint arXiv:1902.06705 (2019).\
> > [3] Athalye, Anish, et al. "Synthesizing Robust Adversarial Examples." International Conference on Machine Learning. 2018.

---

> ### Author Response · Authors · 2020-11-13
> **Response to Reviewer3 [split 1/3]**
>
> We thank the reviewer for taking time to review our paper and their valuable feedback. We understand their concerns and hope that our response will alleviate them. We respond to the each comment one-by-one, below:
>
> > Simply using randomness for adversarial defense, rather than directly improving the robustness of the target model itself (e.g. adversarial training). In the discussion about the Obfuscated Gradient [1], the authors criticized the randomness-based defense method. Although this paper is aimed at defending against black-box attacks, it is still essentially a defense by adding random noise to the input data.
>
> Although the proposed SND adds small random noise to the input, its defense mechanism is quite different from other existing randomization-based defenses in the white-box setting. The randomization-based defenses apply randomization either in networks or input data to make the gradient obtained via back-propagation different from the gradient without randomization. As the reviewer mentioned, Athalye et al. [1] show that these randomization-based defenses can be neutralized through expectation over transform (EOT) technique [2] and criticize them.
>
> On the other hand, SND confuses the gradient estimation and local search of query-based black-box attacks by breaking their assumption that the target model's output exactly corresponds to the query input. EOT approximates the gradient at each gradient descent step by averaging gradients w.r.t. several samples transformed by randomization-based defenses. However, in the black-box setting, the gradients w.r.t. one sample cannot be obtained at once, and the gradients should be estimated using the finite-difference by giving several perturbations. Therefore, even in the process of estimating the gradients w.r.t. the samples, noise is still added to the input, so it will estimate wrong gradients, and the average of the wrong gradients eventually results in erroneous gradients.
>
> Therefore, for a robust attack against SND, it is better for adversaries to lean on more accurate predictions during their gradient estimation and local search by averaging predictions through multiple repetitive inferences for a single image. However, if we average through T queries every time, this leads to using T times as many queries and eventually worsens the query efficiency of black-box attacks. Besides, we find that the attack success rate does not increase significantly, even if we use many queries for expectation. In Table 4, we can observe a trade-off between query efficiency and attack success rate in adaptive attacks. We can also find that even with 100K queries (which is $10\times$ of the original query budget), the attack success rate does not exceed 35%.
>
> [1] Athalye, Anish, Nicholas Carlini, and David A. Wagner. "Obfuscated Gradients Give a False Sense of Security: Circumventing Defenses to Adversarial Examples." ICML. 2018.\
> [2] Athalye, Anish, et al. "Synthesizing Robust Adversarial Examples." International Conference on Machine Learning. 2018.

---

> ### Author Response · Authors · 2020-11-13
> **Response to Reviewer3 [split 2/3]**
>
> > In addition, the proposed method is not useful to defense against white-box attacks and transfer-based attacks, since basically it does not change the predictive model.
>
> In many practical situations, adversaries cannot access the structure and parameters of target models, so they cannot perform white-box attacks against the black-box models. Furthermore, since most defenses for white-box attacks improve their robustness at the cost of loss of clean accuracy, there seems to be a trade-off between robustness against adversarial attacks and clean accuracy, as pointed in [1].
>
> For example, in PGD adversarial training [2], which is well known for effective adversarial defense, the loss of clean accuracy is large as shown in Table 1 in Section 4.2. It drops the clean accuracy (57.9%) by 17.57% compared to that of SND with $\sigma=0.01$ (75.47%). Moreover, the experimental results for the Bandit-TD attack show that the PGD-AT model is weak against the attack as the attack success rate reaches 52.8% when the query budget is 20k, which is much higher than that of SND (8.4%). This implies that existing defenses that are effective against white-box attacks may be weak against query-based black-box attacks.
>
> We think adversarial defenses should have the following goals: (1) preventing malfunction of a model against various attacks, (2) minimizing the computational overhead, (3) maintaining the accuracy on clean images, and (4) easily applicable to existing models. Although SND targets the query-based black-box attacks, it satisfies all the above objectives.
>
> The simple and easy application (it needs just one line of code) of SND and the minimum performance loss of SND allow various service providers to adopt SND for defending their deployed models. Since the architecture and parameters of their model are unknown to attackers, this can effectively defend the model in practical situations.
>
> As the reviewer pointed out, we acknowledge that the proposed method has little effect on defense against white-box attacks and transfer-based attacks. SND is not aimed at defending against white-box attacks, and SND protects models by interfering with gradient estimation and the local search of the query-based black-box attacks. However, for transfer-based black-box attacks, our method is complementary with other defenses, which are mainly effective against transfer-based black-box attacks such as [3, 4]. Combined with these other defenses, it can be robust against general black-box attacks, provided that the model's parameters are kept secret to adversaries.
>
> In our paper, we experimented with Subspace attack [5], which is a powerful hybrid attack exploiting transferability in query-based attacks. The results are shown in Appendix D. Although the attack success rate is high, the reference models of the attacks are ResNet-18 and 34, which are structurally similar to the target model ResNet-50, so transferability is high. When combined with the PGD-AT method, SND shows that it can effectively lower the attack success rate. Judging from these results, thanks to the advantage of SND, which can be easily integrated with other defenses, combined defenses can have robustness against both transfer-based attacks and query-based attacks.
>
> [1] Tsipras, Dimitris, et al. "Robustness May Be at Odds with Accuracy." International Conference on Learning Representations. 2018.\
> [2] Madry, Aleksander, et al. "Towards Deep Learning Models Resistant to Adversarial Attacks." International Conference on Learning Representations. 2018.\
> [3] Tramèr, Florian, et al. "Ensemble Adversarial Training: Attacks and Defenses." International Conference on Learning Representations. 2018.\
> [4] Jalwana, Mohammad AAK, et al. "Orthogonal Deep Models As Defense Against Black-Box Attacks." arXiv preprint arXiv:2006.14856 (2020).\
> [5] Guo, Yiwen, Ziang Yan, and Changshui Zhang. "Subspace Attack: Exploiting Promising Subspaces for Query-Efficient Black-box Attacks." Advances in Neural Information Processing Systems. 2019.

---

> ### Author Response · Authors · 2020-11-13
> **Response to Reviewer3 [split 3/3]**
>
> > Another problem is that while the defense mechanism in this paper is simple, there are equally simple ways to circumvent it. An example is that when the randomness of model output is found by the adversary based on Boundary Attack, the attacker only needs to change the judgment condition of successful attack to “most of the attacks are successful in the same input for multiple queries” to avoid the decrease of attack success rate.
>
> If our understanding of the reviewer’s opinion is correct, we think the main idea of changing the judgment condition of a successful attack is similar to that of the adaptive attack presented in our paper that takes the expectation of multiple queries to get a more accurate prediction for an input.
>
> From the experimental results with the adaptive version of HSJA, which can be seen as an improved version of Boundary attack, SND is difficult to be circumvented with these adaptive attacks. Table 4 shows that it does not increase the attack success rate much even though it consumes many queries, resulting in poor query efficiency. We can develop stronger attacks against SND in the future, but for now, it seems that simple adaptive attacks cannot effectively break SND. Our work shows that we can easily defend query-based black-box attacks with small input noise by breaking the important assumption of them. Interestingly, SND is very simple and easy for defenders, but difficult for attackers to bypass. We hope for the development of powerful attacks that can circumvent SND in the future.

---

> ### Author Response · Authors · 2020-11-23
> **Experimental results of adaptive attacks**
>
> We thank the reviewer's thoughtful concerns. We perform experiments to evaluate the robustness against various adaptive attacks. We conduct adaptive attacks with four decision-based attacks on the CIFAR-10 dataset and two score-based attacks (SimBA-DCT and Bandit-TD) on the ImageNet dataset against SND ($\sigma=0.01$).  These adaptive attacks try to reduce the impact of noise by averaging the model's output using 10 queries for each original query.
>
> | # of queries | 2K$\times$T | 5K$\times$T | 10K$\times$T |
> |-|-|-|-|
> | BA (T=10) | 20.3% | 28.5% | 32.9% |
> | SO (T=10) | 20.5% | 21.1% | 21.9% |
> | HSJA (T=10) | 21.0% | 27.3% | 34.9% |
> | GeoDA (T=10) | 12.1% | 12.2% | 12.2% |
>
> | # of queries | 5K$\times$T | 10K$\times$T | 20K$\times$T |
> |-|-|-|-|
> | SimBA-DCT (T=10) | 8.8% | 9.6% | 11.2% |
> | Bandit-TD (T=10) | 8.8% | 8.8% | 9.2% |
> | SimBA-DCT (T=1) | 8.4% | 8.8% | 10.4% |
> | Bandit-TD (T=1) | 15.2% | 15.2% | 16.4% |
>
> The above results show the robustness of SND against adaptive attacks. In the results, the maximum attack success rate does not exceed 35% even with 100K queries. The above experimental results for adaptive attacks of 4 decision-based attacks and 2 score-based attacks support that SND is difficult to circumvent.

---

### Official Review · AnonReviewer4 · 2020-10-27
**Interesting work on defense against query-based black-box attacks via small random noise**

**Rating:** 6
**Confidence:** 3

**Review:**

#Summary

This paper introduces a simple but effective method by adding a small Gaussian noise to the input, and shows it can neutralize query-based black-box attacks and achieve strong results on both clean accuracy and attack success rates w.r.t. several SOTA attack/defense methods.


#Pros
- The proposed approach is very simple and seems effective across several query-based black-box attacks. The authors have evaluated their SND defense against both decision-based attacks (Sign-OPT, HSJA, GeoDA) and score-based attacks (SimBA, SimBA-DCT, Bandit-TD), and the results are pretty strong in terms of  lowest attack success rates under a fixed query budget, compared with four defense methods (PGD-AT, R&P, RSE, PNI), over both CIFAR and ImageNet.
- The proposed approach achieves good clean accuracy as well, which seems to be another plus compared to existing defenses that achieve strong defense ability but at the cost of clean accuracy.

#Cons
- I think a bit more detailed analysis of how this simple approach works would further strengthen this paper. In Section 3, the authors have provided some example analysis but it would be better to see a more in-depth analysis, e.g., how this method obfuscates the expectation for more general functions in section 3.4; similarly, for section 3.3., the attack objective loss "might increase" but there is no further study provided.

- The following paper adopts a similar idea via additive Gaussian noise to the input for certified adversarial robustness, and should be cited and discussed as well: [1] Bai Li, Changyou Chen, Wenlin Wang, Lawrence Carin. Certified Adversarial Robustness with Additive Noise. NeurIPS 2019.

- As the authors have mentioned, one natural way to attack random-noise based defense is through adaptively querying the model and taking the expectation of T queries, at the cost of increased query budget. In experiments, the authors performed a study of HSJA against SND. Could the authors show more study on this, like how is the query efficiency from adaptive attacks based on other attack methods? Also, since $\sigma$ is fixed, after the first round of queries around a fixed $x$, is there a way to infer what $\sigma$ is and further improves query efficiency in later rounds?

- I think it would also be interesting to further see how this defense works under transfer-based black-box attacks. Does this defense work in general, or only for query-based black-box attacks?


#Overall recommendation

Overall I think the results are strong compared to the baselines and the method is simple and well-explained. The experiments are thorough over various SOTA attacks and defenses, on both CIFAR and ImageNet datasets.  I would vote for accepting the paper.

---

> ### Author Response · Authors · 2020-11-13
> **Response to Reviewer4 [split 1/2]**
>
> We thank the reviewer for their detailed review and constructive feedback that can further improve our paper. We respond to each comment one-by-one, below:
>
> > I think a bit more detailed analysis of how this simple approach works would further strengthen this paper. In Section 3, the authors have provided some example analysis, but it would be better to see a more in-depth analysis, e.g., how this method obfuscates the expectation for more general functions in section 3.4; similarly, for section 3.3., the attack objective loss "might increase" but there is no further study provided.
>
> We tried to support our argument in Section 3.4 with the quadratic function, but we admit that it is structurally different from the network we used. Therefore, we have proved our argument (the error of expectation) with the more general function, a simple network consisting of an affine layer and ReLU activation. We have revised our paper to provide the explanation and proof of this in Section 3.4 and Appendix E. Since deep neural networks have a complex architecture, it may be difficult to provide rigorous mathematical proof, but from our proof on a simple network, we can expect that the average of the output may have an error with the actual output even in a deep neural network.
>
> We apologize for the ambiguity in Section 3.3. Local search-based attacks try to update the image in a direction that decreases the adversarial objective loss. However, since the output is unreliable due to noise, this becomes similar to random motion. Therefore, the attack objective loss might increase in random motion.
>
> >The following paper adopts a similar idea via additive Gaussian noise to the input for certified adversarial robustness, and should be cited and discussed as well: [1] Bai Li, Changyou Chen, Wenlin Wang, Lawrence Carin. Certified Adversarial Robustness with Additive Noise. NeurIPS 2019.
> > [1] Bai Li, Changyou Chen, Wenlin Wang, Lawrence Carin. Certified Adversarial Robustness with Additive Noise. NeurIPS 2019.
>
> We thank the reviewer for their considerate suggestion. We have revised our paper to add a discussion about the paper in Section 5 and highlighted the added and modified sentences.

---

> ### Author Response · Authors · 2020-11-13
> **Response to Reviewer4 [split 2/2]**
>
> > As the authors have mentioned, one natural way to attack random-noise based defense is through adaptively querying the model and taking the expectation of T queries, at the cost of increased query budget. In experiments, the authors performed a study of HSJA against SND. Could the authors show more study on this, like how is the query efficiency from adaptive attacks based on other attack methods?
>
> Following the constructive suggestion, we evaluate adaptive attacks against other attack methods other than HSJA. The following table shows the attack success rates of adaptive versions of 4 different attacks that take expectation through 10 repetitive queries for each query (T=10) against SND ($\sigma=0.01$) on the CIFAR-10 dataset.
>
>
> | # of queries | 2K$\times$T | 5K$\times$T | 10K$\times$T |
> |-|-|-|-|
> | BA (T=10) | 20.3% | 28.5% | 32.9% |
> | SO (T=10) | 20.5% | 21.1% | 21.9% |
> | HSJA (T=10) | 21.0% | 27.3% | 34.9% |
> | GeoDA (T=10) | 12.1% | 12.2% | 12.2% |
>
> In the results, HSJA shows the best query efficiency in most cases, but the attack success rate does not exceed 35% even with 100k queries. We also revised our paper to add the above table as Table 6.
>
>
> > Also, since σ  is fixed, after the first round of queries around a fixed x, is there a way to infer what σ is and further improves query efficiency in later rounds?
>
> It is difficult for adversaries to estimate the $\sigma$ of SND because they cannot access the actual input image with random noise, but they can only obtain its output. If we assume that the attackers know the $\sigma$ used by SND, it can adaptively determine how many queries to use to get the expectation for query efficiency. In general, if $\sigma$ is high, then the changes in outputs will also be large.
>
> > I think it would also be interesting to further see how this defense works under transfer-based black-box attacks. Does this defense work in general, or only for query-based black-box attacks?
>
> SND protects models by interfering with gradient estimation and the local search of query-based attacks. However, we do not expect SND is effective against transfer-based attacks as these attacks exploit the transferability of adversarial examples. However, our method is complementary with other defenses, which are mainly effective against transfer-based black-box attacks such as [1, 2]. Combined with these other defense techniques, it can work well against general black-box attacks, provided that the model's parameters are kept secret to adversaries. For example, in our paper, we experimented with Subspace attack [3], which is a powerful hybrid attack that exploits transferability in query-based attacks. The results are shown in Appendix D.
>
> Because the reference models of the attacks are ResNet-18 and 34, which are structurally similar to the target model ResNet-50, the transferability of gradients is high, so the attack success rate is high in the case of Baseline, SND(0.01) and SND(0.001). When combined with the PGD-AT, SND shows that it can effectively lower the attack success rate (61.2% $\rightarrow$ 42.4%).
>
> [1] Tramèr, Florian, et al. "Ensemble Adversarial Training: Attacks and Defenses." International Conference on Learning Representations. 2018.\
> [2] Jalwana, Mohammad AAK, et al. "Orthogonal Deep Models As Defense Against Black-Box Attacks." arXiv preprint arXiv:2006.14856 (2020).\
> [3] Guo, Yiwen, Ziang Yan, and Changshui Zhang. "Subspace Attack: Exploiting Promising Subspaces for Query-Efficient Black-box Attacks." Advances in Neural Information Processing Systems. 2019.

---

> ### Author Response · Authors · 2020-11-23
> **Additional results of the adaptive attacks**
>
> Following the reviewer’s invaluable advice, we also conduct adaptive attacks with two score-based attacks (SimBA-DCT and Bandit-TD) with T=10 on the ImageNet dataset against SND ($\sigma=0.01$). The experimental results show that these adaptive attacks cannot effectively circumvent SND.
>
> | # of queries | 5K$\times$T | 10K$\times$T | 20K$\times$T |
> |-|-|-|-|
> | SimBA-DCT (T=10) | 8.8% | 9.6% | 11.2% |
> | Bandit-TD (T=10) | 8.8% | 8.8% | 9.2% |
> | SimBA-DCT (T=1) | 8.4% | 8.8% | 10.4% |
> | Bandit-TD (T=1) | 15.2% | 15.2% | 16.4% |

---

### Official Review · AnonReviewer1 · 2020-10-28
**The paper proposes a new defense against adversarial examples created through query based attacks. The defense is based on adding a small amount of random noise. The authors provide some theoretical arguments and experimental evaluation of their approach.**

**Rating:** 4
**Confidence:** 5

**Review:**

The paper proposes a new defense against adversarial examples created through query based attacks. The defense is based on adding a small amount of random noise. The authors provide some theoretical arguments and experimental evaluation of their approach.

Overall evaluation
While the presented idea might be promising neither the theoretical nor the experimental evaluation are sufficient to validate this claim. The research on adversarial robustness is very active and many attacks and defenses have been proposed. To ensure good quality of research, proposals have been made on how to correctly evaluate a proposed defense (see below). The paper falls short of these standards. The main shortcoming is the lack of a robust evaluation against an adaptive attack (i.e. an attacker aware of the defense). The majority of the analysis in the paper is restricted to evaluating  the defense against existing attacks, robustness against an attacker that does not take the defense into account is insufficient.
The paper provides a short analysis of an adaptive attack, however, this new proposed attack is just a slight variation of an existing attack. This type of defense against a "patched" attack is still insufficient. Especially because from the paper it is unclear why this specific attack was chosen, if it is optimal and if it addresses the shortcomings of the original attack. One clear oversight is that the objective of the attacker changes, as it is not looking for a point that gets misclassified, but for a point that gets misclassified even under noise.
The shortcoming of the analysis goes beyond the lack described above and I recommend the authors to go through the given guidelines.

References
Carlini, Nicholas, et al. "On evaluating adversarial robustness." arXiv preprint arXiv:1902.06705 (2019).
Athalye, Anish, Nicholas Carlini, and David Wagner. "Obfuscated gradients give a false sense of security: Circumventing defenses to adversarial examples." arXiv preprint arXiv:1802.00420 (2018).

Concrete comments about the unclarity
"the above observation implies that clean images have a relatively long distance to the decision boundary" -- the existence and prevalence of adversarial examples implies exactly the opposite of this statement, if this would be true there would not be any adversarial examples.

Section 3.2 is very unclear. eta which is defined in Section 3.1. a independent of $x$ is suddenly dependent on $x$. The sentence "this function is very noisy" is just a claim. It is unclear why the random function becomes discontinuous for large Var.

Section 3.3 is just saying that existing attacks don't take into account the newly proposed defense.

The analysis in Section 3.4 is for a quadratic function. Most current CNN architectures don't have any quadratic component and in fact the Resnet architecture used in the experiments is a piecewise linear function with a final softmax layer. I fail to see the relevance of the presented argument.

**After Rebuttal**

I thank the authors for their detailed remarks and clarification. While it is encouraging to see that the authors have offered some clarifications and assurances regarding the validity of their approach, my main concern is whether the edit distance between the current work and the proposed modifications is just too high. It appears to me that there simply is too much that the authors need to modify in order to obtain an acceptable manuscript (with the other reviewers' concerns and suggestions as well). I believe that the paper can be significantly improved if the authors incorporate the comments from the current round of reviewing.

---

> ### Author Response · Authors · 2020-11-13
> **Response to Reviewer1 [split 1/2]**
>
> We thank the reviewer for their detailed review and their constructive feedback. We understand their concerns and hope that our response will alleviate them. We respond to each comment one-by-one, below:
>
> > The main shortcoming is the lack of a robust evaluation against an adaptive attack (i.e. an attacker aware of the defense). The paper provides a short analysis of an adaptive attack, however, this new proposed attack is just a slight variation of an existing attack. Especially because from the paper it is unclear why this specific attack was chosen, if it is optimal and if it addresses the shortcomings of the original attack. One clear oversight is that the objective of the attacker changes, as it is not looking for a point that gets misclassified, but for a point that gets misclassified even under noise.
>
> We totally agree with the reviewer’s comment that solid research for adversarial defense requires testing against adaptive attacks that exactly know the working principle of the defense. As the reviewer pointed out, Athalye et al. [1] show that randomization-based defense such as R&P used in our paper can be circumvented through the expectation over transform (EOT) technique [2]. Considering the suggestion of [1] and [3], we design the adaptive attack against SND, which adapts the EOT technique.
>
> EOT computes the gradients by averaging the gradient w.r.t. random samples as follows.
>
> $\nabla E_{t\sim T}f(t(x))=E_{t\sim T}\nabla f(t(x))$
>
> For the case of SND, $t(x)=x+\eta,$ where $\eta \sim N(0,\sigma^2)$.
>
> In the white-box setting, $\nabla f(t(x))$ can be directly obtained via back-propagation because all parameters of the model are known. However, in the black-box setting, since adversaries cannot access the parameters of the model, $\nabla f(t(x))$ should be estimated with finite-difference via methods such as RGF described in Section 3.2. However, the adversary does not know the sampled $\eta$ in $t(x)$ and the model noise also causes error in the estimation of $\nabla f(t(x))$. Therefore, it is better to adapt EOT in the computation of finite-difference by taking the expectation with multiple queries to estimate the exact prediction for an image.
>
> To complement the robust evaluation against an adaptive attack, we conduct additional experiments with attack methods other than HSJA. The following table shows the attack success rates of adaptive versions of 4 different attacks that take expectation through 10 repetitive queries for each query (T=10) against SND ($\sigma=0.01$) on the CIFAR-10 dataset.
>
> | # of queries | 2K$\times$T | 5K$\times$T | 10K$\times$T |
> |-|-|-|-|
> | BA (T=10) | 20.3% | 28.5% | 32.9% |
> | SO (T=10) | 20.5% | 21.1% | 21.9% |
> | HSJA (T=10) | 21.0% | 27.3% | 34.9% |
> | GeoDA (T=10) | 12.1% | 12.2% | 12.2% |
>
> The above results show the robustness of SND against adaptive attacks. In the results, the maximum attack success rate does not exceed 35% even with 100k queries. We can observe a trade-off between query efficiency and attack success rate in adaptive attacks. For query-based black-box attacks, we should also consider the query efficiency, so it is difficult to develop an efficient attack that evades SND.
>
> As another form of adaptive attack against SND, we can increase the size of random perturbation in the gradient estimation method to give perturbation large enough to ignore the disturbance of small input noise. However, increasing the perturbation size can enlarge the gradient estimation error by itself.
>
> Although it is difficult to devise effective adaptive attacks against SND, we can make our defense even stronger. Considering that the mean of the small input noise is zero, It can be a potential vulnerability of SND. So, we can introduce $m$ that is sampled from $N(0,\sigma^2)$ every $t$ minutes. If $m$ is added to the image with $\eta$ (i.e., $t(x)=x+\eta+m$), the average over a finite period of time becomes nonzero. We can also combine R&P and SND to enhance their defensive ability further.
>
> We agree that the presented adaptive attack is not an optimal attack against SND. However, the development of sophisticated attacks to break SND seems to be beyond the scope of our paper because the aim of our paper is to highlight the effectiveness of small input noise against query-based black-box attacks. Through our paper, we want to inform that future studies on query-based black-box attacks need to evaluate the robustness against models with SND.
>
> References\
> [1] Athalye, Anish, Nicholas Carlini, and David Wagner. "Obfuscated gradients give a false sense of security: Circumventing defenses to adversarial examples." arXiv preprint arXiv:1802.00420 (2018).\
> [2] Athalye, Anish, et al. "Synthesizing robust adversarial examples." International conference on machine learning. PMLR, 2018.\
> [3] Carlini, Nicholas, et al. "On evaluating adversarial robustness." arXiv preprint arXiv:1902.06705 (2019).

---

> ### Author Response · Authors · 2020-11-13
> **Response to Reviewer1 [split 2/2]**
>
> > Concrete comments about the unclarity "the above observation implies that clean images have a relatively long distance to the decision boundary" -- the existence and prevalence of adversarial examples implies exactly the opposite of this statement, if this would be true there would not be any adversarial examples.
>
>
> We apologize for the confusion in the statement. From the statement, we want to say that clean images are relatively farther to the decision boundary than the adversarial examples of query-based black-box attacks. We acknowledge that adversarial examples do not need to be near the decision boundaries as long as it successfully fools the model. However, due to the nature of most query-based black-box attacks that try to keep the prediction of the adversarial class while minimizing the perturbation norm, they are usually located near decision boundaries.
>
> Table 5, in our paper, supports our argument. It can be seen that the probability of prediction changes with additive Gaussian noise is significantly higher in the query images during adversarial attacks than in clean images. Although many correctly classified images are included in the calculation of the probability, the correctly classified query images can cross the decision boundaries with a little perturbation. So the distance between the adversarial images and them is small, and we can think that adversarial examples are more vulnerable to additive Gaussian noise than clean images. Gilmer et al. [1] establish close connections between the adversarial robustness and corruption robustness with additive Gaussian noise. Moreover, the paper [1] says “Finally, this suggests that methods which increase the distance to the decision boundary should also improve robustness to Gaussian noise, and vice versa.” Therefore, we conjecture that clean images are relatively farther to the decision boundary than adversarial examples as clean images are more robust against additive Gaussian noise.
>
> [1] Gilmer, Justin, et al. "Adversarial examples are a natural consequence of test error in noise." International Conference on Machine Learning. 2019.
>
> > Section 3.2 is very unclear. eta which is defined in Section 3.1. a independent of x is suddenly dependent on x. The sentence "this function is very noisy" is just a claim. It is unclear why the random function (?!) l* becomes discontinuous for large Var.
>
> We apologize for the confusion in Section 3.2. Considering that $\eta$ in Section 3.1 is newly sampled at each query, we can see that $\eta$ eventually depends on $x$ and $t$. Even if we fix  $t=T$, we have different samples of $\eta$ according to $x$, and to indicate this, we denote $\eta(x)$ for $\eta$ in Section 3.2. Since $\eta(x)$ is sampled differently depending on x, If $\eta(x)$ is added to x, $f(x+\eta(x))$ becomes a discontinuous function for $x$.
>
> Simplifying the problem, if $f$ is a one-dimensional function $R \rightarrow R$ and sampled $\eta(1)=0.06$ and $\eta(1.0001)=-0.03$ then, $f(1)$ becomes $f(1.06)$ and $ f(1.0001)$ becomes $f(0.9701)$. Therefore, if the variance of $f$ is large, the sample function becomes more noisy and discontinuous.
>
> > Section 3.3 is just saying that existing attacks don't take into account the newly proposed defense.
>
> We apologize for the confusion in Section 3.3. In section 3.3, we explain that existing local search-based attacks have difficulty in finding the adversarial examples. But, in section 3.4, we describe an adaptive attack strategy that considers the proposed defense.
>
> > The analysis in Section 3.4 is for a quadratic function. Most current CNN architectures don't have any quadratic component and in fact the Resnet architecture used in the experiments is a piecewise linear function with a final softmax layer. I fail to see the relevance of the presented argument.
>
> We tried to support our argument in Section 3.4 with the quadratic function, but we admit that it is structurally different from the network we used. Therefore, we have proved our argument (the error of expectation) with the more general function, a simple network consisting of an affine layer and ReLU activation. We have revised our paper to explain and prove this in Section 3.4 and Appendix E. Since deep neural networks have a complex architecture, it may be challenging to provide rigorous mathematical proof, but from our proof on a simple network, we can expect that the average of the output may have an error with the actual output even in a deep neural network. The empirical results of Table 4 support our argument.

---

> ### Author Response · Authors · 2020-11-23
> **Additional experimental results of adaptive attacks**
>
> We thank the reviewer's thoughtful concerns.
> We perform additional experiments to evaluate the robustness against adaptive attacks.
> We conduct adaptive attacks with two score-based attacks (SimBA-DCT and Bandit-TD) with T=10 on the ImageNet dataset against SND ($\sigma=0.01$). The experimental results show that these adaptive attacks cannot effectively circumvent SND.
>
> | # of queries | 5K$\times$T | 10K$\times$T | 20K$\times$T |
> |-|-|-|-|
> | SimBA-DCT (T=10) | 8.8% | 9.6% | 11.2% |
> | Bandit-TD (T=10) | 8.8% | 8.8% | 9.2% |
> | SimBA-DCT (T=1) | 8.4% | 8.8% | 10.4% |
> | Bandit-TD (T=1) | 15.2% | 15.2% | 16.4% |
>
> The experimental results for adaptive attacks of 4 decision-based attacks and 2 score-based attacks support that SND is difficult to bypass.

---

### Official Review · AnonReviewer2 · 2020-10-30
**A simple yet effective idea for defending against adversarial query-based black-box attacks on deep neural networks.**

**Rating:** 7
**Confidence:** 3

**Review:**

In this paper, the authors propose a novel method for defending against
adversarial attacks on deep neural networks in a black-box setting. The idea is
simple as it is effective: to simply add some small Gaussian noise to the input
prior to passing it through the model. The authors make some heuristic arguments
for why this might be effective, including the difficulty of estimating
gradients (for gradient-based attacks) and robustness against local-search based
attacks that do not account for variability in the model output. They also
present a set of experiments demonstrating the superiority of their defense
against a variety of common attack techniques, and contrast the performance of
Small Noise Defense (SND) against other common defense techniques.  The
experiments demonstrate that SND maintains a higher clean accuracy (i.e.,
accuracy on unperturbed, non-adversarial images) compared to other defense
methods, while also staying robust to a variety of attack techniques.

Strengths of the paper:
  + The performance of the approach, given its simplicity (as the authors
    point out, it only requires one line of code!), is striking and makes
    SND an attractive option. It will likely be widely adopted given its ease
    of implementation, in situations where protection against adversarial
    attacks is critical.
  + The experimental evaluation was rigorous and supports the paper's main
    claims.
  + The paper is well organized, well written and easy to follow.

Areas for improvement/questions for the authors:
  - The theoretical analysis of the SND model, given its simplicity, is somewhat
    lacking. The authors themselves acknowledge this in their conclusion, and
    cite this as a promising future avenue to explore. Currently, the paper's
    arguments are mostly empirical.
  - Given the similarity to Random Smoothing (Sec. 5), I was curious as to why
    that method was not included in the experimental evaluation. Is this because
    RS uses large variance noise, and is thus unlikely to maintain a sufficient
    clean accuracy? Some clarification here would be welcome.

On balance, I think this paper presents some interesting results, and it's rare
to see such simple ideas have such startling payoffs. That alone makes me
positive about this paper. While the theoretical work is somewhat light, I don't
see that as fatal -- I think this paper as is opens up some interesting avenues
for future work, that could strengthen this contribution even further. I thus
vote for ACCEPTANCE.

---

> ### Author Response · Authors · 2020-11-13
> **Response to Reviewer2**
>
> We thank the reviewer for their positive feedback and their encouraging comment. We acknowledge the main argument of the current paper is currently somewhat empirical, but we believe that further research into theoretical work will be enriched in the future.
>
> Below, we respond to their constructive comment on the comparison with randomized smoothing.
> Randomized smoothing is a certified defense method which trains models with Gaussian data augmentation. The resultant smoothed classifiers have certified radius on the norm-bounded adversarial perturbations. Whereas, SND does not require to change the training method, and it only affects the input image in test time to defend against query-based attacks. Since the working principle and purpose of the two methods are quite different, we did not include randomized smoothing in the experimental evaluation. Nevertheless, we agree that it would be informative to compare the defensive ability of the randomized smoothing method. Following the kind suggestion, we evaluate the robustness of the pretrained smoothed classifier (ResNet-50 trained on ImageNet dataset) downloaded from the official GitHub(<https://github.com/locuslab/smoothing>) of the authors of RS against HSJA.  For fair comparison, we also use the baseline model (RS with $\sigma=0$) trained by the authors of RS. The attack success rates are evaluated as follows.
>
> | # of queries | 5K | 10K | 20K |
> |-|:-:|:-:|:-:|
> | Baseline | 66.0% | 85.2% | 98.0% |
> | Randomized smoothing ($\sigma$=0.5) | 79.2% | 83.6% | 86.0% |
>
> The results show that the smoothed classifier is extremely vulnerable to HSJA. Interestingly, when the query budget is 5K, RS is weaker than the baseline. Compared to RS, SND significantly reduces the attack success rate at 20K queries (86.0% $\rightarrow$ 8.4%).

---

### Author Response · Authors · 2020-11-11
**We uploaded the code**

We thank all the reviewers for their constructive and detailed feedback.
Although the actual code for SND is just one line, we uploaded our experimental code for reproduction.
We are conducting various experiments according to the thoughtful feedback.
After we organize our results and thoughts, we will respond to the invaluable comments within a few days.

---

### Decision · Program_Chairs · 2021-01-07
**Final Decision**

**Decision:**

Reject

**Comment:**

The paper proposes a defense against black-box adversarial example attacks based on adding small Gaussian noise to the inputs. Its evaluation is carried out empirically using CIFAR-10 and ImageNet datasets.

Despite a somewhat complete experimental evaluation (on two datasets) the lack of theoretical justification strongly affects the significance of the proposed method. It can be clearly seen from the experimental results that the proposed level of noise is a trade-off between clean accuracy and attack effectiveness. However, this tradeoff neither implies a substantial degree of security (the attack success rate is roughly halved but this does imply robustness against attacks since the initial success rate is rather high) nor is  the impact to clean accuracy negligible. Furthermore, the robustness of the proposed method against an attack which is aware of such defense (similar to the Kerckhoff's principle in cryptography) is not evaluated. The authors mentioned several directions for addressing this issue in their response but implementation of such improvements is impossible within the level of revisions acceptable in a post-review process.

A major revision of the paper taking into account the feedback provided by the current reviews would certainly improve its acceptance chances.